# Anti-UV Microgel Based on Interfacial Polymerization to Decrease Skin Irritation of High Permeability UV Absorber Ethylhexyl Methoxycinnamate

**DOI:** 10.3390/gels10030177

**Published:** 2024-03-02

**Authors:** Wei Wang, Qi-Tong He, Yin-Feng Chen, Bai-Hui Wang, Wen-Ying Xu, Qing-Lei Liu, Hui-Min Liu

**Affiliations:** 1School of Perfume & Aroma and Cosmetics, Shanghai Institute of Technology, Shanghai 201418, China; wangweittg@sit.edu.cn (W.W.); aliceho1001@163.com (Q.-T.H.); cqcyf@outlook.com (Y.-F.C.); wbh0506@126.com (B.-H.W.); 216072132@mail.sit.edu.cn (W.-Y.X.); liuqinglei@sit.edu.cn (Q.-L.L.); 2Engineering Research Center of Perfume & Aroma and Cosmetics, Ministry of Education, Shanghai 201418, China

**Keywords:** polymerization, microgel, skin irritation, sunscreen, EHMC

## Abstract

Ethylhexyl methoxycinnamate (EHMC) is frequently employed as a photoprotective agent in sunscreen formulations. EHMC has been found to potentially contribute to health complications as a result of its propensity to produce irritation and permeate the skin. A microgel carrier, consisting of poly(ethylene glycol dimethacrylate) (pEDGMA), was synthesized using interfacial polymerization with the aim of reducing the irritation and penetration of EHMC. The thermogravimetric analysis (TGA) indicated that the EHMC content accounted for 75.72% of the total composition. Additionally, the scanning electron microscopy (SEM) images depicted the microgel as exhibiting a spherical morphology. In this study, the loading of EHMC was demonstrated through FTIR and contact angle tests. The UV resistance, penetration, and skin irritation of the EHMC-pEDGMA microgel were additionally assessed. The investigation revealed that the novel sunscreen compound, characterized by limited dermal absorption, had no irritant effects and offered sufficient protection against ultraviolet radiation.

## 1. Introduction

Lipophilic UV agents are more likely to penetrate the skin because of their lipid structure. A recent study indicated that three common UV filters all showed detection in plasma or urine. In addition, the increasing skin temperature exposed to the sun increases the probability of sunscreen agent penetration [1].

The lipophilic chemical known as ethylhexyl methoxycinnamate (EHMC) is commonly utilized in sunscreen formulations used in daily life to improve the effectiveness in protecting against ultraviolet (UV) radiation. Nevertheless, EHMC has garnered much attention and debate in relation to its possible implications for health [2,3,4]. EHMC was detected in the blood and urine of both rat and human models using high resolution quadrupole time-of-flight mass spectrometry (Q-TOF-MS). Furthermore, it was determined that there were five metabolites present in the blood samples [5]. During an in vitro study, it was revealed that both trans- and cis-EHMC exert genotoxic effects on HL1-hT1, which are a type of adult human liver stem cells. The findings of the Comet assay indicated that exposure to high amounts of cis-EHMC led to considerable DNA damage, reaching a classification of level 4 [6]. Previous studies on EHMC have shown evidence supporting its ability to enhance the proliferation of inflammatory cytokines, notably tumor necrosis factor-α (TNF-α) and interleukin-6 (IL-6). In addition, it was revealed that the expression of IL-6 mRNA exhibited a significant rise of more than 15% [7].

The number of studies focused on mitigating the permeability or irritation of EHMC has been on the rise in response to the consistent discoveries of possible hazards to human health. The technique of microencapsulation has been referenced in pertinent academic research as a means to mitigate the adverse impacts of EHMC on human health. Perugini et al. described the fabrication of nanoparticles loaded with EHMC, which exhibited enhanced photostability through the utilization of salt fractionation [8]. Scalia et al. demonstrated that the lipid microparticles (LMs) loaded with EHMC exhibited reduced permeability compared to the unencapsulated EHMC in the in vivo skin penetration assay [9]. In their study, Wu et al. used poly(methyl methacrylate) (PMMA) as the coating material for the fabrication of EHMC microcapsules. These microcapsules were shown to have a modest level of transdermal permeability and demonstrated consistent resistance against ultraviolet (UV) radiation [10]. A subsequent study conducted by the same research team documented the development of SiO_2_-encapsulated nanoparticles using the sol–gel emulsion technique, resulting in a cumulative release decrease of EHMC exceeding 60% [11]. In recent years, there has been a growing body of evidence supporting the efficacy and viability of ultrasound-assisted technology in the formation of core–shell microcapsules of EHMC [12]. The utilization of ecologically benign materials as the outer shell of microcapsules is highly advantageous owing to their inherent biocompatibility and biodegradability properties. Lignin, PMMA, and poly (ethylene glycol) (PLA) are biocompatible polymers that can be used to transport drugs. In drug loading investigations, lignin encapsulation of target medicines is sustainable and environmentally friendly. These days, lignin-based encapsulation is used for enhancing EHMC performance. Li et al. researched into how lignin models and EHMC performed in terms of UV protection. The outcome showed that even after three hours of UV irradiation, the sunscreens made of lignin models and EHMC maintained good photostability. The sunscreens containing 4-coumaric acid and EHMC showed better UV-shielding effectiveness in 16 lignin models, with a SPF of 19.37. The improvement was attributed to the interaction between the aromatic rings of the lignin models and EHMC through PD-stacked π–π stacking [13]. 

The pertinent research provided useful insights; nevertheless, the majority of these new findings were primarily centered on minimizing the infiltration of EHMC, rather than addressing both dangers, namely skin irritation and skin permeation. The fundamental criteria for selecting components include favorable biocompatibility and effective cross-linking performance. The preparation of hydrogels based on pEDGMA was achieved by combining the ultrasound-assisted technique with polymerization [14]. The utilization of pEDGMA as a loading ingredient was deemed appropriate owing to its favorable biocompatibility [15]. Consequently, it has been extensively employed in various studies involving drug carriers [16,17,18,19] and degradable tissue [20]. This application is based on the polymerization [21,22] of ethylene glycol dimethacrylate (EDGMA) through covalent cross-linking, specifically radical polymerization [23,24,25]. In a recent study conducted by Suhail et al., interpenetrating polymer network (SIPN) hydrogels were synthesized using the cross-linker EDGMA via the free radical polymerization technique. This approach effectively regulated the release of ibuprofen [14]. In an effort to enhance the transfection effectiveness of pullulan, Caroline et al. [26], employed the aza-Michael addition process [27] to adorn pullulan with EGDMA. In order to enhance the extraction efficiency of caffeic acid from wine, Elhachem et al. employed N-phenylacrylamide, ethylene glycol dimethacrylate, and azobisisobutyronitrile as the monomer, cross-linker, and initiator, respectively, to synthesize a functional molecule using radical polymerization [28].

In this study, an approach was undertaken to mitigate the irritation caused by EHMC by encapsulating it with pEDGMA generated using radical self-polymerization. Notably, this particular combination has not been previously documented in the available research. Both homopolymerization [29] and the microemulsion technique, which are ecologically friendly experimental procedures, were used in the synthesis of the EHMC-pEGDMA microgel.

## 2. Results and Discussion

### 2.1. The Characterization of EHMC-pEDGMA Microgel

#### 2.1.1. The SEM of EHMC-pEDGMA Microgel

The SEM analysis in Figure 1 revealed that the EHMC-pEDGMA microgel exhibited an irregularly spherical shape. Furthermore, the surface of the microgel seemed smooth, indicating that there would be no severe friction between the material and the skin. The majority of materials produced through polymerization following emulsion had a spherical shape and possessed a smooth surface. This distinctive trait was likely attributed to the emulsion process, as indicated by previous studies [30,31,32]. Prior to the initiation of EDGMA polymerization using the APS solution, the oil-in-water emulsion exhibited a uniformly spherical morphology. Subsequently, the formation of pEDGMA occurred at the interface between oil and water, resulting in the ultimate configuration of a globular molecule. Furthermore, the absence of fissures and hollows on the material surface depicted in the image suggests that the coating created by pEDGMA had remained intact. The presence of an undamaged coating had a direct impact on stability as it guaranteed the absence of any leakage of internal components from the materials. The short black bar indicates 0.5 μm.

#### 2.1.2. The PSD Measurement of EHMC-pEDGMA Microgel

The particle size of the EHMC-pEDGMA microgel is a significant indicator of material properties (Figure 2). The findings indicated that the average particle size measured 517.3 nm, aligning closely with the SEM test result of approximately 500 nm. The UV absorbance span of anti-UV agents was found to be directly controlled by the particle size of the material in previous research studies [33,34,35]. The reduced particle size confers advantages in terms of enhanced UVB absorbance. Conversely, the strong resistance to ultraviolet A (UVA) radiation is associated with a larger particle size. While sunscreen with exceptional UVA resistance might offer a high sun protection factor (SPF), the presence of large particle sizes often results in an undesirable, visibly white appearance. In practice, it is advisable to maintain a particle size that closely aligns with that of typical sunscreen emulsions, as this approach allows for a harmonious blend of natural aesthetics and a high sun protection factor (SPF). The reported nanoemulsions exhibited a particle size distribution ranging from 100 to 1000 nm. This similar characteristic indicated the natural dispersion of the EHMC-pEDGMA microgel in aqueous solutions, resulting in a suspension with a color resembling that of sunscreen emulsions.

#### 2.1.3. FTIR Spectroscope

The FTIR analysis conducted using the tableting method revealed that the EHMC-pEDGMA microgel powder, which was synthesized by the freeze-drying procedure, exhibited susceptibility to breakage when subjected to severe pressure (Figure 3). In this instance, the characteristic absorption of EHMC was effectively detected; it was seen at 2940 cm^−1^, which corresponds to the stretching vibration of the C-H bond. Additionally, a stretching vibration of the ester group C=O bond was observed at 1708 cm^−1^. The stretching vibration peak of the C=C bond in the aromatic compound was observed at 1614 cm^−1^, whereas the bending vibrational peak of the C-H bond in the aromatic compound was observed at 828 cm^−1^. These spectral findings suggest that the benzene ring in EHMC is para-substituted. The absorption peaks seen in the FTIR spectrum of the pEGDMA polymer corresponded to specific vibrational modes. Specifically, the peak at 2940 cm^−1^ was attributed to the stretching vibration of C-H bonds, the peak at 1739 cm^−1^ was associated with the stretching vibration of the ester group C=O bond, and the peak at 1165 cm^−1^ was indicative of the stretching vibration of the C-O bond within the pEGDMA polymer. The spectra of the EHMC-pEDGMA microgel exhibited distinct peaks corresponding to various vibrational modes. Specifically, the stretching vibration peak of C-H at 2945 cm^−1^ and the ester C=O peak at 1720 cm^−1^ were observed. Additionally, the stretching vibration peak of C=C in aromatic compounds at 1614 cm^−1^, the C-O-C peak in aromatic ethers at 1255 cm^−1^ and 1165 cm^−1^, as well as the bending vibrational peaks of C-H at 829 cm^−1^ in aromatic compounds were also detected. It may be inferred that the microgel was mainly composed of two components: EHMC and pEDGMA.

#### 2.1.4. Contact Angle Test

The contact angle test was conducted to evaluate the stability and wetting property of the EHMC-pEDGMA microgel suspension generated using deionized water and gas condensate, respectively. The contact angle of deionized water was measured to be 44 ± 1°, whereas the contact angle of the microgel dispersion was found to be less than 90° at 45 ± 1°. These results indicate that the surface of the sunscreen microgel is hydrophilic, demonstrating good wetting properties. The contact angle of the gas condensate was measured to be 17 ± 1°. However, upon the introduction of the EHMC-pEDGMA microgel, the contact angle increased to 27 ± 2°. This observation suggested that the stability of the EHMC-pEDGMA microgel was insufficient when utilized in the context of oils or lipids. The observed contact angle exhibited an approximate increase of 10°, which could perhaps be attributed to the presence of surfactant PVA emanating from the microgel [36].

#### 2.1.5. The Thermogravimetric Analysis of the Sunscreen Microgel

The curves depicted in Figure 4 exhibit weight loss at two distinct temperatures, indicating the capacity to differentiate between the core ingredient and the shell materials. This differentiation allows for the evaluation of loading content and the assessment of thermal stability [37]. Within the temperature range of 100–250 °C, it was observed that the sample exhibited a notable escalation in weight loss, particularly at 130 °C, which closely aligns with the boiling point of EHMC. At a temperature of 212 °C, the rate of weight loss reaches its maximum, with a corresponding loss ratio of around 75.72% in this particular range. The pEGDMA exhibited significant weight reduction within the temperature range of 350–500 °C, with the highest rate of weight loss observed at 338 °C. The thermogravimetric analysis (TGA) results revealed that EHMC constituted the largest fraction of the microgel. Furthermore, the sample exhibited negligible weight loss below 100 °C, indicating that the microgel had satisfactory thermal stability.

### 2.2. The Performance of EHMC-pEDGMA Microgel

#### 2.2.1. Skin Penetration

In comparison to the original material, EHMC, the findings from the skin permeation test of the EHMC-pEDGMA microgel demonstrated its ability to reduce the penetration of EHMC in human skin. As depicted in Figure 5, there was a progressive rise in the cumulative penetration of EHMC over the course of time. The error bars in both groups displayed normal values, with measurements falling within a range of 0.15 μg/cm^2^. In the initial hour, the measured quantities of EHMC in the respective receiving chamber were insufficiently distinct to ascertain the outcome, likely due to testing deviation resulting from operational and equipment factors. Following a duration of 12 h, the control group exhibited an approximate accumulation of 18.26 μg/cm^2^ of EHMC, whereas the sunscreen microgel demonstrated a result of 4.60 μg/cm^2^. Despite the relatively high error levels observed in the EHMC-pEDGMA group, there was a considerable divergence between the EHMC group and the EHMC-pEDGMA group. The decrease in penetration of EHMC-pEGDMA was attributed to the macromolecular nature of the polymer pEDGMA, which was characterized by large particle sizes. This characteristic posed challenges for the infiltration of pEDGMA into the stratum corneum and pores, resulting in a noticeable reduction in penetration efficiency [38,39,40].

#### 2.2.2. Skin Irritation Assays

The results shown in Figure 6 proved that the sunscreen agent EHMC exhibited a moderate level of irritation, whereas the EHMC-pEGDMA microgel demonstrated a non-irritating effect. This observation provided evidence supporting the notion that EHMC possessed favorable biocompatibility and could be considered safe for application on human skin.

The results presented in Table 1 and Figure 7 demonstrate that the positive control exhibited irritant effects, such as bleeding, vascular lysis, and clotting in the group treated with a 0.1 mol/L NaOH solution (Figure 7b). The observation of capillary bleeding and vascular lysis in the image suggested that the EHMC diluted with C12-15 alkylbenzoate at a concentration of 1% had the potential to cause harm to human skin. In order to specifically attribute this outcome to the EHMC, a solvent control (Figure 7c) was included in the subsequent procedure, which demonstrated the absence of any irritating reaction. The suspension of the EHMC-pEDGMA microgel was subjected to testing in order to assess the integrity of the chorioallantoic membrane. Specifically, the objective was to determine whether the membrane exhibited any signs of bleeding, vascular lysis, or clotting when exposed to varying concentrations of the microgel suspension. The findings that were examined and analyzed revealed that the primary determinant of the absence of an irritating reaction was the compatibility of pEDGMA. Another valid reason was that the surface of the EHMC-pEGDMA microgel exhibited no remaining EHMC.

#### 2.2.3. Sun Protection Factor

Figure 8 displays the SPF values obtained for two sunscreen samples, namely the EHMC group and the EHMC-pEDGMA group. The error bars for both groups show a range of variation within one, which can be considered an acceptable observation given that the sun protection factor was not less than five. The UV protection efficacy of the EHMC-pEDGMA microgel was shown to be significantly superior to that of the original UVB absorbent EHMC. The determination of the sun protection factor is influenced by various factors, such as the methodology employed during experimentation, the formulation used, and the specific apparatus utilized for measurement [41,42]. The primary distinction observed in both the composition and functionality of the sunscreen recipe was the variation in the anti-UV ingredient utilized. The EHMC-pEDGMA microgel exhibited the ability to disperse effectively in the aqueous phase, hence offering a notable benefit in their application inside oil-in-water emulsions. One notable observation is that the particle size of the EHMC-pEDGMA microgel was found to be lower compared to the typical particle size of emulsions. This characteristic was advantageous in the formation of a sunscreen film with closely scattered sunscreen agent [43]. The experimental findings additionally indicated that the inclusion of pEDGMA exhibited a favorable influence on the EHMC anti-UV capability. The encapsulating approach described herein may have potential applicability to other frequently employed UV absorbents.

## 3. Conclusions

The EHMC-pEDGMA microgel was synthesized using radical polymerization assisted by microemulsion technology. Although the preparation was not any easier than for the emulsion formulation and traditional lipid microparticles, the coating based on polymerization was more stable. Additionally, the EHMC-pEDGMA microgel technique proved efficient and safe for the environment. When compared to a lengthy reaction preparation process, the preparation might be completed in as little as 4 h, saving electricity. This microgel met the necessary criteria, which included being non-toxic and non-irritating, exhibiting appropriate UV absorption, demonstrating minimal penetration, and possessing outstanding thermal stability. The material’s non-toxic characteristic was essential and served as a benchmark for choosing the correct part to use as the shell. Therefore, the majority of findings in the research on EHMC may select for the cytotoxicity test or neglect the non-toxic testing of the finished product. Some materials, however, had difficulty dissolving in water, which led to test inaccuracies and prevented the material from being able to be analyzed. The HET-CAM test was appropriate in this study to demonstrate the EHMC-pEDGMA microgel’s non-irritating characteristics and safety. Due to the HET-CAM test’s lack of a dissolution limit and the membrane’s obvious reactions, it was possible to determine the material’s direct irritating and hazardous state. The scarcity of these sunscreen microgels posed a constraint on cost-effective production. The necessity of exploring and optimizing the preparation of microgels for industrial applications was evident. Furthermore, the EHMC-pEDGMA microgel’s improved performance suggested that the other anti-UV agents with minor harmful and irritating flaws would have the possibility to be upgraded.

## 4. Materials and Methods

### 4.1. Materials

Ethylhexyl methoxycinnamate (EHMC), polyvinyl alcohol (PVA), ethylene glycol dimethacrylate (EDGMA), ammonium peroxydisulfate (APS), sodium metabisulfite (Na_2_S_2_O_5_), capric triglyceride (GTCC), potassium bromide, monometallic sodium orthophosphate, disodium hydrogen phosphate, sodium chloride, and sodium hydroxide, C12-15-alkylesters were all AR (99.7% purity) and purchased from China Shanghai Titan Technology Co., Ltd. (Shanghai China). The fresh pig skin used in the Transdermal absorption test was from China Shandong Zhifu Rong biological studio (Shandong China).

### 4.2. Methods

#### 4.2.1. Preparation of EHMC-pEDGMA Microgel

10 g PVA powder and 90 g water were added to a beaker, heated up to 95 °C and stirred to prepare a 10 wt% PVA solution as an emulsifier. 15 g 10 wt% PVA solution and 1 g deionized water was mixed to form the aqueous phase. The oil phase was prepared by adding 27 g EHMC and 3 g GTCC, followed by the addition of 4 g the monomer EDGMA. The aqueous phase was then added to the oil phase while being homogenized. To achieve better homogeneity and stability, the emulsion was transferred to the ultrasonic cell disruption system. After ultrasonic disrupting with 450 W for 30 min, an oil-in-water emulsion was obtained.

The ratio of APS as the initiator and Na_2_S_2_O_5_ as the catalyst was 1:1, accounting for 0.4% in the entire reactant mixture. The preliminary emulsion was introduced to a three-neck flask and purged with N_2_ at 30 °C for 30 min, then the initiator was added and allowed to react for an additional 5 min. In the oil phase, the olefinic bonds of monomer EDGMA were broken by catalyst in the aqueous phase, and next, the polymerization reaction was triggered to form pEDGMA [44,45]. After adding the catalyst to the reaction mixture, the reaction was processed within 3 h. The microgel was finally stored in the fridge, and the dry microgel was obtained by the freeze-drying process. 

#### 4.2.2. Morphological Observation and Particle Size

The morphology of the EHMC-pEDGMA microgel was characterized by a scanning electron microscope (SEM, S-3400N, Hitachi, Tokyo, Japan) under dry conditions. The particle size distribution (PSD) of the EHMC-pEDGMA microgel dispersed in the deionized water was analyzed by the laser particle size analyzer (LS, Nano-ZS, Malvern Limited, Fareham, UK).

#### 4.2.3. Fourier Infrared Spectroscopy (FTIR) Analysis

The infrared spectra of pEGDMA, EHMC, and EHMC-pEDGMA microgel were measured to observe their components by the Fourier infrared spectrometer (FTIR, VERTEX 70, Bruker Limited, Berlin, Germany).

#### 4.2.4. Contact Angle Analysis

The wetting performance of the EHMC-pEDGMA microgel was determined by measuring the contact angle. The static contact angles of the EHMC-pEDGMA microgel dispersions, dispersed by deionized water and gas condensate, respectively, were measured in the experiments performed using a glass sheet as the solid substrate. Gas condensate is a liquid consisting of straight short-chain alkanes, which is also a frequently used agent to substitute common oil in contact angle tests.

#### 4.2.5. Thermogravimetric Analysis

The composition of the sunscreen microgel was determined by the thermal gravimetric analyzer (TGA, Q-5000, TA Limited, New Castle, DE, USA). The behavior of microgel weight loss was observed at a heating rate of 20 °C/min in the temperature range of 20–600 °C.

#### 4.2.6. Skin Permeation Behavior

In this study, the researchers employed fresh pig skin sourced from the dorsal region of a Bama miniature pig as a substitute for human skin in the Franz test. The methodology employed in this study was derived from relevant research [46] and subsequently adjusted to align with the actual conditions of the test. A mixture of ethanol and normal saline was prepared in a 1:1 volume ratio to serve as the receptor medium in the receiving chamber (20 mL) of the Franz diffusion tank. A dispersion of sunscreen microgel at a concentration of 1 mg/mL was generated as the donor vehicle. A pig skin sample, shaped like a coin and with an approximate size of 4.90 cm^2^, was positioned between the donor and receptor. The precise measurement of the area that was involved in the experiment was around 2.54 cm^2^. The receptor medium was maintained at a temperature of 25 °C and agitated at a speed of 500 revolutions per minute (rpm). The receptor medium was sampled at regular intervals of time, with a volume of 1 mL being extracted for the purpose of measuring UV absorbance at a wavelength of 310 nm. To preserve the initial volume, 1 mL of receptor media was introduced. The standard curve approach was employed to determine the concentration of EHMC in various samples. The equation of the cumulative penetration quality was as follows:(1)$$Q_{t}=\frac{V_{t}C_{t}+\sum_{t-1}V_{i}C_{i}}{A}.$$

*C_t_* and *C_i_* represented the concentration obtained at the *t*th and the (*t* − 1) sampling points, respectively. *V_t_*, *V_i_*, and A each represented the volume of the receiving chamber, the sampling volume, and the effective penetration area, respectively.

#### 4.2.7. Skin Irritation Assays

Skin irritation was investigated by the cosmetics ocular irritant and the corrosive HET-CAM tests derived from the industry standard [47]. The SPF fertilized embryo (Zhejiang Lihua Agricultural Science and Technology Co., Ltd., Zhejiang, China) was selected and their eligible chicken embryos on the ninth day of hatching were ready. In this assay, six chicken embryos from each sample group, one negative control, one positive control, one solvent control, and a reference control were set. The reaction time method with irritation score (IS) is generally used for the transparent liquid test objects. The experimental controls included a positive control (0.1 mol/L NaOH solution), a negative control (NaCl solution with a mass concentration of 0.9%), a solvent control (C12-15 alkyl benzoate, deionized water), and a reference control (1% EHMC diluted with C12-15 alkylbenzoate). Once the 0.3 mL sample was added to the membrane, the moments referring to hemorrhage, vascular lysis and clotting were recorded in the scheduled 5 min. The result was calculated by the *IS* formula equation as follows:(2)$$IS=\frac{\left(301-\sec H\right)\times 5}{300}+\frac{\left(301-\sec L\right)\times 7}{300}+\frac{\left(301-\sec C\right)\times 9}{300}.$$

Sec *H*, Sec *L*, and Sec *C* represent the average time (seconds) at which hemorrhage, vascular lysis, and clotting occurred, respectively.

The EHMC-pEDGMA microgel dispersion was outside the standard of the IS method. During the determination of the end point score (ES), a total of 0.3 mL dispersion was exposed to the entire membrane surface. After 3 min of exposure, the dispersion was carefully rinsed with the configured 0.9% NaCl solution for 30 s. The result was evaluated using the irritation score. The score corresponds to no reaction, mild, moderate, and severe bleeding due to three reactions, including hemorrhage, vascular lysis, and clotting. The ES was obtained from the average total score. For the IS evaluation, an IS lower than 1 indicates no irritant, an IS between 1 and 5 indicates a light irritant, an IS between 5 and 9 indicates a moderate irritant, and an IS higher than 10 indicates corrosive. In the ES evaluation, the average total score under 12 is rated as no or light irritant, between 12 and 16 is moderate irritant, and beyond 16 is corrosive.

#### 4.2.8. Sun Protection Factor

Sun protection factor (SPF) is a symbol of sunscreen performance, and in this study, the SPF of the EHMC group and the EHMC-pEDGMA group were tested by a sunscreen analyzer (SPF-290AS, Solar Light Limited, Glenside, PA, USA). The sunscreen formula of the two groups was mostly the same, except for the addition of 5% anti-UV agent EHMC or EHMC-pEDGMA. The dosage of each sample was about 0.3 g; the two samples were both applied to the PMMA plates and then placed in a dark place. After 15 min, two samples were prepared to be analyzed. Each sample was tested in three parallel experiments.

## Figures and Tables

**Figure 1 gels-10-00177-f001:**
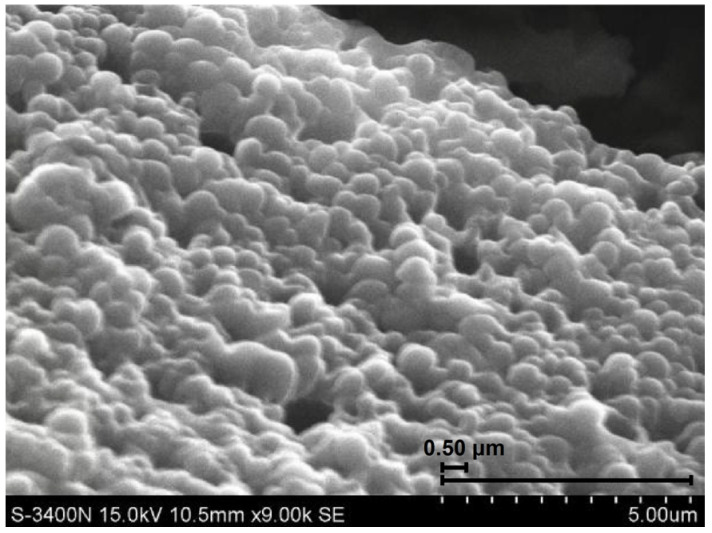
The SEM image of EHMC-pEDGMA microgel.

**Figure 2 gels-10-00177-f002:**
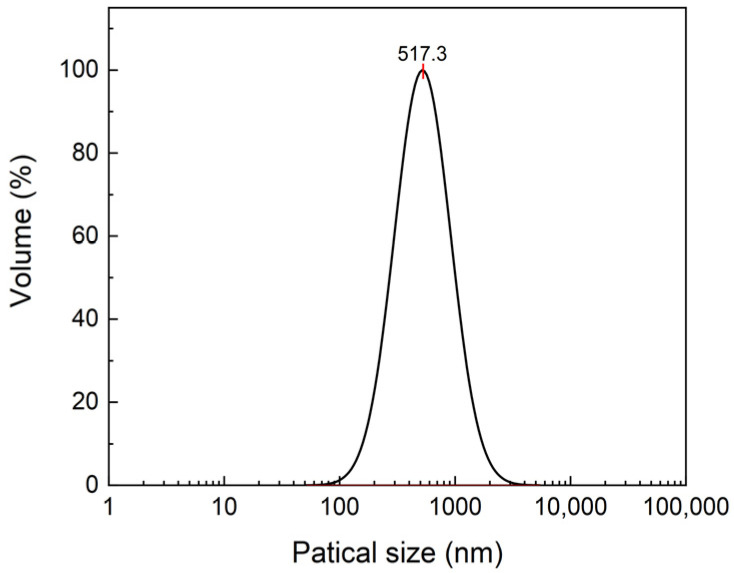
The PSD graph of EHMC-pEDGMA microgel.

**Figure 3 gels-10-00177-f003:**
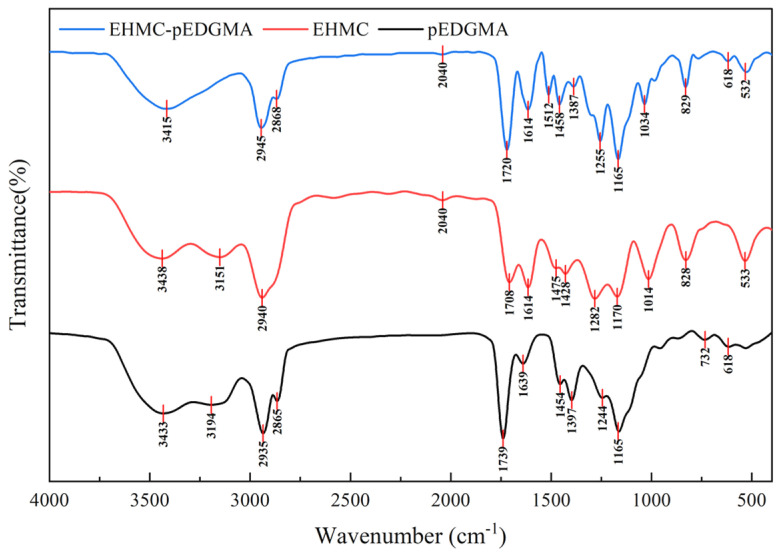
The FTIR graph of EHMC-pEDGMA, EHMC, and pEDGMA.

**Figure 4 gels-10-00177-f004:**
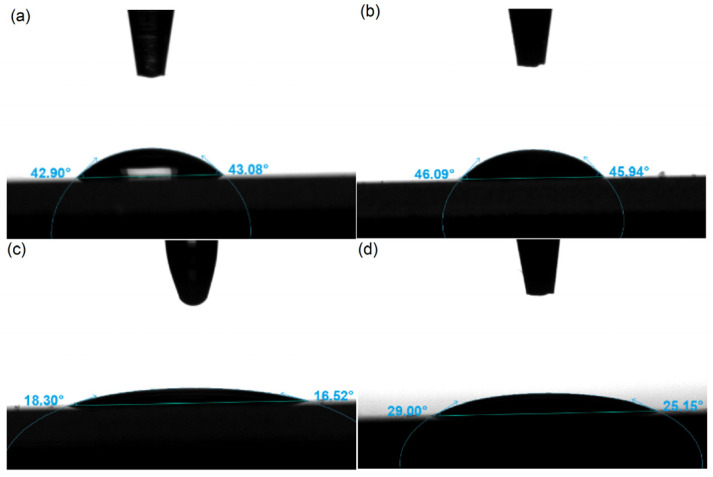
Contact angle determination between different media: (**a**) deionized water; (**b**) aqueous dispersion of EHMC-pEDGMA microgel; (**c**) gas condensate; (**d**) gas condensate with EHMC-pEDGMA microgel.

**Figure 5 gels-10-00177-f005:**
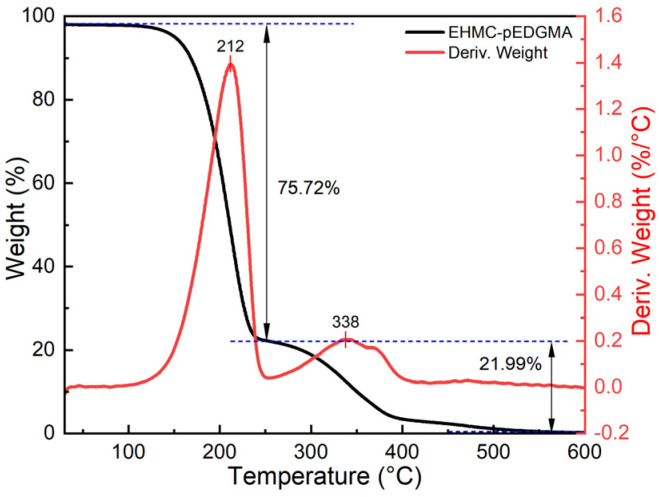
Thermogravimetric analysis of EHMC-pEDGMA microgel.

**Figure 6 gels-10-00177-f006:**
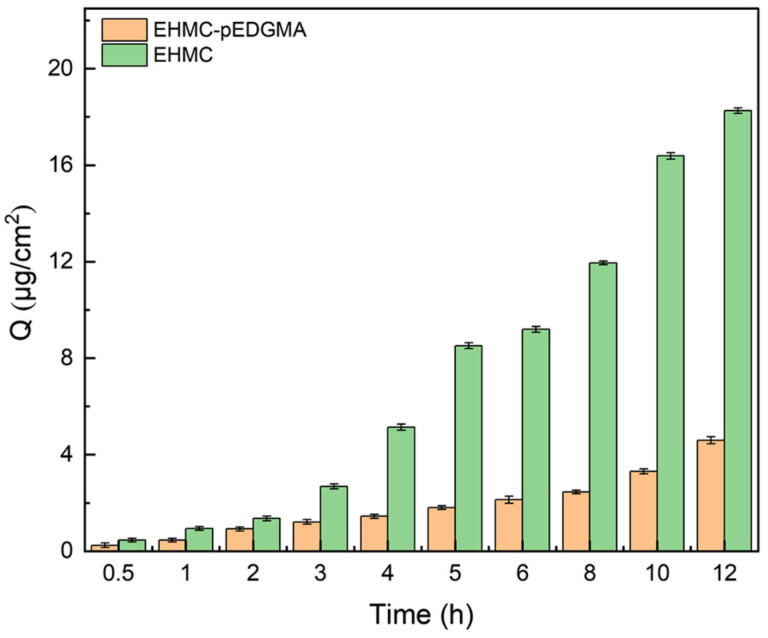
Skin penetration behavior of EHMC and EHMC-pEGDMA microgel.

**Figure 7 gels-10-00177-f007:**
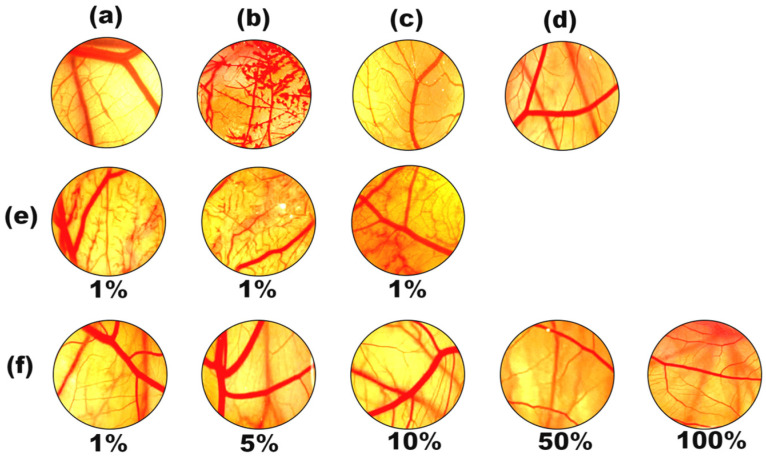
The blood vessels of chick embryo chorioallantoic membrane after treatment with different test substances and their enlarged images: (**a**) 0.9% *w*/*v* NaCl solution; (**b**) 0.1 mol/L NaOH solution; (**c**) C12-15 Alkyl Benzoate; (**d**) Deionized water; (**e**) three parallel samples of 1% EHMC solution; (**f**) 1–100% *w*/*v* EHMC-pEGDMA microgel dispersion.

**Figure 8 gels-10-00177-f008:**
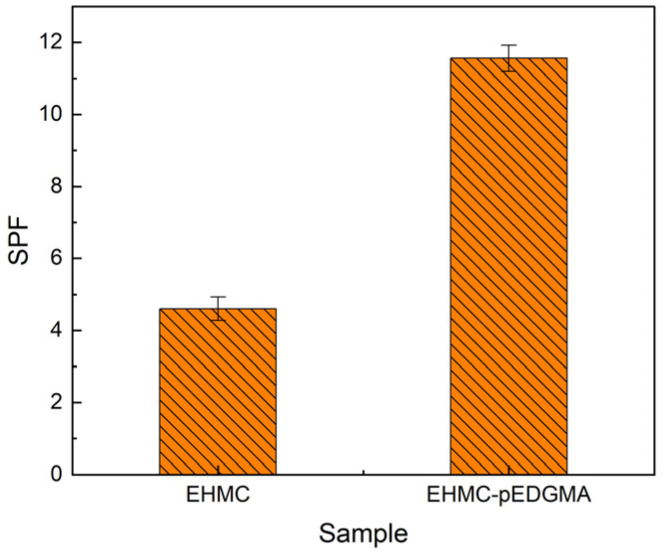
The SPF of EHMC group and EHMC-pEDGMA group.

**Table 1 gels-10-00177-t001:** Test results for chicken embryo chorioallantoic membrane.

Sample	Concentration	IS	ES	Irritant
EHMC-pEDGMA microgel	1%	-	0	no irritant
	5%	-	0	no irritant
	10%	-	0	no irritant
	50%	-	0	no irritant
	100%	-	0	no irritant
EHMC (diluted with C12-15 alkylbenzoate)	1%	5.42	-	moderate irritant
0.1 mol/L NaOH solution	positive control	17.21	-	corrosive
0.9% NaCl solution	negative control	0	-	no irritant
C12-15 alkyl benzoate	solvent control	0	-	no irritant
Deionized water	solvent control	0	-	no irritant

## Data Availability

The data presented in this study are openly available in the article.

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
