# Peer review of "Anti-UV Microgel Based on Interfacial Polymerization to Decrease Skin Irritation of High Permeability UV Absorber Ethylhexyl Methoxycinnamate"

_gels, 2024, doi:10.3390/gels10030177_

Round 1

Reviewer 1 Report

Comments and Suggestions for Authors

I recommend major revision since many sentences are very difficult to understand. For example, lines 111-112: what is "scalic" vibration?

The cited works, in the Introduction (Line 44 to 53), could be described in more details followed by the scientific issue that the present paper is addressing,  based on the gaps left by previous works. 

The use o contact angle measurements to assess stability is not explain, nor is informed what is "condensate".

In short, the paper is potentially interesting, however, since it is poorly written and the results poorly explained/discussed, I would not recommend its publication in its present form. 

Comments on the Quality of English Language

The paper needs major revision. 

Author Response

Thank you for your valuable comments! Please see the attachment.

Reviewer 2 Report

Comments and Suggestions for Authors

The authors examined possibility of encapsulating common anti-UV agent (ethylhexyl methoxycinnamate - EHMC) using nanogels carrier composed of poly(ethylene glycol dimethacrylate) (pEDGMA) formed by interfacial polymerization in order to reduce health issues related to its application such as skin damage and systemic absorption. The authors reported lower penetration and no irritation comparing to EHMC itself.  However, manuscript is poorly written, hard to read (due to poor sentence construction and English language)  and results are not discussed well. R&D part is only briefly presenting obtained results and there are several subsections without a single reference. Obtained results should be presented more clearly and properly discussed, including comparison with relevant literature data to demonstrate both scientific and practical relevance of the study. Several specific remark:

Last sentence of Abstract should be rephrased, better anti-UV ability compared to what?

Page 1, line 35 and page 2, line 48: Did authors mean In an in vitro study…. caused…? and …EHMC in the in vivo skin…

Page 2, line 73: It is written: From above relative researches, what does it mean? Similar phrases that doesn't sound right could be found throughout the manuscript. 

Page 2, lines 79-81: These are results and, therefore, should not be part of the introduction section.

For the skin irritability determination, sources according to which analyses were done should be referenced.

Manuscript is missing subsection with explanation of statistical analyses and software used. Also, legend of each figure should contain data regarding number of independent experiments and meaning of error bars.

Comments on the Quality of English Language

Extensive editing of English language required.

Author Response

(The authors gave the same response as above.)

Reviewer 3 Report

Comments and Suggestions for Authors

Point 1:- Zeta potential of particles in nanogel is not mentioned 

Point 2:- Skin penetration section you mentioned fresh pig skin , kindly mention which part of skin was taken

Point 3:- mention the sources of pig skin vendor etc

Point 4:- In line 251-252 ,  Ethanol and 251 normal saline were mixed at a volume ratio of 1:1 as the receptor medium. Why the buffers are not used instead of ethanol and saline. justify the same

Author Response

(The authors gave the same response as above.)

Reviewer 4 Report

Comments and Suggestions for Authors

I have completed the review of the manuscript titled "Anti-UV Nanogel Based on Interfacial Polymerization to Decrease Skin Irritation of High Permeability UV Absorber Ethylhexyl Methoxycinnamate," submitted by [Author's Name] et al. I find the work to be generally novel; however, some revisions are recommended to enhance the overall quality of the manuscript.

  1. Introduction Section:

    • Address the importance of references related to the stabilization of Ethylhexyl Methoxycinnamate (EHM) using microcapsules. The use of ecologically benign materials, such as lignin nanostructures, for EHM stabilization should be highlighted, citing the work of Piccinino et al., 2022.
  2. Conclusion Section:

    • Extend the conclusion to emphasize the valuable results obtained throughout the manuscript. Discuss the implications of the findings in the context of the broader field and potential applications.
    • Compare the results obtained in this study with previous research in the same domain. Highlight similarities, differences, and advancements, providing a comprehensive overview for readers.
Comments on the Quality of English Language

No comments

Round 2

Reviewer 1 Report

Comments and Suggestions for Authors

The paper still needs English revision: for example "A nanogels carrier".

About the word "nanogels", I have two comments, first, as I understant, only one formulation was prepared, so the correct term would be "nanogel" (singular); second, the term "nano" does not apply correctly since many of the objects (too many) seen in Fig 1 and 2 are above 1 micrometer. The correct would be to refer to micro and sub-micro nanogel.

About the experimental section: line 276 "certain amount of deionized" - what is the amount?

Line 302 " dispersed by deionized water and gas condensate" - I still do not know what is the gas condensate (composition, etc). 

Comments on the Quality of English Language

Minor editing of English language required
